# Assessing parent-child interaction with deaf and hard of hearing infants aged 0–3 years: An international multi-professional e-Delphi

**Martina Curtin** [1,2]*, **Madeline Cruice**[2◉], **Gary Morgan**[3◉], **Rosalind Herman**[2◉]

**1** Speech and Language Therapy (Paediatrics, Community), Homerton Healthcare NHS Foundation Trust, London, United Kingdom, **2** Language and Communication Science, City, University of London, London, United Kingdom, **3** Universitat Oberta de Catalunya, Psychology and Education Sciences, Barcelona, Spain

◉ These authors contributed equally to this work.
* martina.curtin.1@city.ac.uk

**Data Availability Statement:** All data are available in the City, University of London Figshare repository at https://doi.org/10.25383/city.25188572.v12.v1

## Abstract

### Introduction

Most deaf babies are born to hearing families who do not yet have the skills to communicate effectively with their child. Adaptations to communication are important because the quality of parent-child interaction (PCI) predicts how a deaf child develops language. Teachers of Deaf children and Speech and Language Therapists support families with communication in the home. Currently, there are no assessments that appraise how a parent interacts with their deaf baby. Previous research has identified which parent behaviours and approaches are used in PCI assessments in research and practice. The current paper forms consensus on the core content and best practices of a new PCI tool for deaf children aged 0–3 years.

### Methods

An international sample of expert academics and practitioners (n = 83) were recruited to take part in a two-round modified electronic Delphi study. Participants were presented with 69 statements focusing on (i) which parent behaviours were important in assessment (ii) the methods to be used in PCI assessment. Participants rated the extent to which they agreed or disagreed with each statement on a five-point Likert scale and gave comments to support their response. Consensus was defined as ≥80% of participants rating the statement as a (4) 'highly important' or a (5) 'essential'. If consensus was not reached, participant comments were used to generate new statements which were rated in the second round. This project involved a patient and public involvement (PPI) group of hearing and deaf parents and professionals to design and guide the study.

### Results

Consensus was achieved on 52 statements and ranged from 80–99%. A further six statements were additionally included. Within the 58 statements included, 36 were parent behaviours which centred on the parent's observation of, and response to, their child's behaviour and/or language. The remaining 22 statements focused on methods used in the assessment

**Funding:** This study was funded by the National Institute for Health Research (NIHR) as part of Martina Curtin's Clinical Doctorate Research Fellowship (NIHR Research Trainees Coordinating Centre, NIHR300558). The funders had no role in study design, data collection and analysis, decision to publish, or preparation of the manuscript. The views expressed are those of the authors and not necessarily those of the NIHR or the Department of Health and Social Care.

**Competing interests:** The authors have declared that no competing interests exist.

such as parents having their PCI filmed, parents having the opportunity to review the video and assess themselves alongside a professional, and parents being involved in subsequent goal setting.

## Conclusions

This e-Delphi presented the parent behaviours and methods of assessment to be included in a new PCI tool for deaf children. Future co-production work and acceptability and feasibility testing are discussed.

## Introduction

Deaf children are most often born to hearing families [1] who have not yet experienced deafness and therefore require support with adapting their communication skills [2]. In the current study, 'parent' refers to the mother, father, or any primary caregiver and 'deaf' refers to all deaf and hard of hearing children, identified with a mild, moderate, severe, and/or profound level of deafness.

The quality of parent-child interaction (PCI) predicts deaf children´s language development [3,4]. A recent systematic review reported that the quality of parents' linguistic input explained 31% of the variance in deaf children's spoken language scores [5]. In many studies, deaf children's spoken language development is 1–1.5 standard deviations lower than hearing peers [6]. Children with severe and profound degrees of deafness are the most delayed [7]. Much research suggests that communication is one of the major sources of stress when parenting a deaf child [8,9]. There is an internationally accepted guideline of working with families of deaf babies, known as '1-3-6 model' where hearing screening occurs before the infant is 1 month old, a deafness diagnosis is complete by 3 months, and intervention is provided by 6 months [10]. Professionals (such as Teachers of Deaf children (ToDs), Speech and Language Therapists (SLTs), and Psychologists) maximise on this early start by offering family-centred support, education, and coaching. Professionals might suggest adaptations to a parent's communication approach that includes greater focus on gaining the attention of the deaf infant, or on maintaining the joint attention between parent and child [11]. A parent may need support with ensuring their interactions are accessible and perceivable by the child; this might include the introduction of a signed language. There is evidence that by providing parents with the knowledge, skills and practice they need to enhance their communicative behaviours, positive changes in deaf children's communication are observed [11,12].

Guidance for professionals working within early intervention programs for deaf children recommends the provision of tailored, individualised support to families [13]. An important first step to being able to offer this support is comprehensive assessment. Whilst there are assessments to track the deaf child's communication skills, an assessment tool to appraise a parent's strengths and needs in communicating with their deaf child does not yet exist. A recent systematic review of 61 international studies [14] identified which parent behaviours and assessment methods are most frequently used by researchers of PCI with deaf children aged 0–3 years. The parent skills identified were attention getting, joint engagement, parental sensitivity, and facilitation techniques that either enriched the deaf child's language or ensured good access to the parent's language input (ibid). In addition, the systemic review reported that many of these parent behaviours were associated with positive child language outcomes. Researchers tended to film mothers (rather than fathers), in labs for an average of 19 minutes.

These videos were analysed frame by frame, with in-depth and time-consuming coding systems. Some of the studies used scales for measuring PCI instead of, or in addition to, coding. It is important to know the evidence-based parent behaviours positively associated with deaf children's language development, however it is not straightforward to use these methods in clinical practice.

A recent survey of 190 UK-based professionals working with deaf infants aged 0–3 years [15] reported that PCI assessments overlapped with parent skills highlighted in the aforementioned systematic review [14]. In addition, a further 18 practice-based parent behaviours not reported in the systematic review were identified. In contrast to researchers, however, professionals often observed PCI live and made mental or paper-based notes, using their own knowledge and skills to analyse the interactions. Most professionals also stated that their assessments always or often led to collaborative goal planning with parents. In a sequential, focus group study, professionals shared how complex and multi-faceted assessing PCI in families' homes can be [16]. Without any published guidance on how to conduct PCI assessments, it is difficult for professionals to know which parent behaviours are core, which to prioritise within their PCI observations, and which assessment approaches are best particularly when working within busy, complex family systems. Additionally, professionals do not have a formal way of showcasing progress to parents (or managers) through re-assessment. Therefore, an e-Delphi consensus study was designed to refine and gain expert international consensus on perceived best practice for an assessment of PCI where the child is deaf and aged 0–3 years. The objectives were to: 1) reach consensus on which parent behaviours to include in a new assessment and 2) develop best practice recommendations on how to approach assessing these behaviours.

The Delphi technique is an anonymous, iterative, and multistage method to synthesise expert opinion into a group consensus. It is carried out using questionnaires and feedback over a series of rounds [17]. The anonymity of participants aims to avoid a small number of experts dominating the discussion or the peer pressure to conform, and instead means full participation from all participants involved [18]. Delphi studies are used in health sciences to identify priorities [19], in policy making and developing practice guidelines [20], and where 'best practice' agreement is desired [21]. The current e-Delphi builds on previous research by the same authors [14–16].

## Methods

Ethical approval was granted from City, University of London's School of Health & Psychological Sciences Research Ethics Committee (ETH2122-0790). The reporting guideline for Delphi studies [22] has been used.

### E-Delphi study design and modifications

This study employed a modified, two-round, electronic, international Delphi methodology to investigate the aims outlined previously. Delphi studies traditionally utilise two to three rounds of controlled feedback [23]. Typically, the first round is used to generate ideas, opinions, and issues via literature searching and open-ended questions. Previous work [14–16] meant the aggregation of ideas in a 'classic' round 1 Delphi was obsolete. Additionally, whilst some Delphi studies use face to face and/or online meetings, an online survey format was used to facilitate optimum access to international experts.

### Statement generation and number of statements

Previous work [14–16] produced a set of parent behaviours and assessment approaches which were converted into statements. A parent involvement group (see PPI section below) also

generated four statements that were relevant and meaningful to them. Final statements were reviewed and agreed by the remaining members of the authorship team.

## Pilot testing

The e-Delphi was trialled on four professionals (a QToD, an SLT, and two academics) working in the field of deafness. The piloting process was helpful in considering the usability and technical functioning of the e-Delphi. For example, following the pilot, a 'back' button was introduced to allow ratings to be changed. In addition, it was noted that once a sliding scale had been clicked on, the cursor was automatically set on 'not essential' unless moved by the participant. This had two potential consequences: it may have been difficult for participants to see they had missed a statement, and also, an incorrect rating may have been submitted. To avoid this, the original sliding scale was removed and replaced with a multiple-choice layout with the following five options: no importance (1), low importance (2), important, but not essential (3), high importance (4), and essential (5). Pilot testing also assisted with amendments to wording and ensured the terminology was appropriate for both practitioners and academics, e.g., the statement 'Parent is genuinely interested and involved' was improved by adding the sentence 'In research this is called availability'. Likewise, examples were given for statements, e.g., for 'Parent using auditory attention-getting strategies' the following example was added for clarity: 'e.g., using the child's name'.

## Definition of an expert

The use of experts in Delphi panels are fundamental to reliability [24]. Experts ensure the outcomes of an e-Delphi have content, face, and concurrent validity [25,26]. In the current study, included experts were defined as:

- An academic who has a paper included in the review related to the assessment of parent-child interaction with deaf children 0–3 years [14] or an author on the best practice principles for family-centered early intervention paper [13] and/or

- A qualified professional who self-reported to have 10 years (or more) of experience working with deaf children. This included SLTs, ToDs, Psychologists, Psychiatrists, Mental Health Practitioners (in the UK these were based at National Deaf Child and Adolescent Mental Health Services - NDCAMHS).

   Turoff and Linstone [27] recommend including between 10 and 50 experts in a Delphi study. This is similar to a recent overview (review of systematic reviews) of e-Delphis [23], where the median number was 40 (but ranged from 3 to 731). Keeney, Hasson, and McKenna [28] argue that the expert sample size is dependent on the area of focus, the complexity of the problem, the heterogeneity of the sample and availability of resources. The current study included 102 experts from within the field of early years deafness and parent-child interaction as a larger, international sample helps with generalisability.

## Recruitment and inclusion criteria

Each academic participant was sent an invitation to register via email. Professional participants were recruited by three different approaches: 1) The registration link for the e-Delphi was promoted within a range of UK professional networks and clinical excellence groups; 2) Emails were sent to professionals who had participated in the UK survey [15] who had registered their interest and fit the eligibility criteria, and 3) Academics were asked to forward information about the study to any eligible practitioners within their city or country.

## Consent and registration process

To take part in the e-Delphi, participants needed to click on a registration link embedded in their invitation email. On the link's first page, a downloadable information sheet explained the study's purpose, the name and contact details of the investigator, the approximated completion time of each survey round, and information on data storage. On the second page, participants gave their consent to participate by ticking a series of tick box questions, including consent for their answers to be used in an anonymized and aggregated manner to derive consensus statements. On the third page, participants gave their name, email address, professional background, hearing status, the city and country where they were based, and the number of years of experience they had within the field. This information was only available to the first author to enable round-to-round survey monitoring. Participants were invited to register from 21st March until the 11th May 2022. Participation was voluntary, and no incentives were offered.

## Response and participation rates

Qualtrics identified each participant as a unique visitor through their IP address. There were 102 unique visitors to the information and registration page. One hundred and two (102) people agreed to participate by clicking the consent boxes. Ninety-five (95) participants continued to the registration form to leave their demographic information. Each registrant was sent their own individualised link to begin the e-Delphi. Out of the 95 participants who had registered, 83 completed round 1.

Qualtrics was used for both e-Delphi rounds. E-Delphi round 1 was open from 20th April– 13th May 2022 (23 days) and e-Delphi round 2 was open from 6th June to 30th June 2022 (24 days). For each round, participants were given two prompts and a final reminder to encourage completion. Failure to respond was considered participant attrition.

## Procedure

For both e-Delphi rounds, participants were asked to consider the content and format of a new PCI assessment for deaf 0–3 year olds and their caregivers and rate each statement using a 5-point scale of no importance (1), low importance (2), important, but not essential (3), high importance (4), and essential (5). For all items in both rounds, the rating of each statement was mandatory. Participants were able to change their answers using a 'back' button. An optional open-text box was provided per item for participants wishing to explain their rating.

## Consensus

A-priori criteria were established using an adapted GRADE system [29]. The GRADE system uses clear cut and transparent rules for rating using concrete categories such as 'high', 'moderate' and 'low'. Eighty percent (80%) of e-Delphi respondents needed to rate a statement as a 4 ('high importance') or 5 ('essential') for consensus to be achieved and for the statement to be added to the proposed assessment tool. Likewise, 80% of participants needed to rate a statement as a 1 ('no importance') or 2 ('low importance') for it to be eliminated from the next round. Statements achieving consensus also needed an IQR of <1.

## Round 1

In round 1 of the e-Delphi, there were a total of 69 statements for participants to rate across two sections. First, there were 40 parent behaviours (PBs) to rate across five categories: attention getting (number of statements = 5); joint engagement (n = 4); parental sensitivity (n = 8); increasing access to language (n = 4); and language enrichment (n = 19). Each category was

presented on a new page of the survey and each item within that category was presented in a random order to reduce order bias (the influence of the previous statement on the response to subsequent statements). After rating the 40 parent behaviours, participants were asked to list any missing parent behaviours in an open text box. In the second section of round 1, there were 29 statements related to the approach professionals should take to assess parents (abbreviated to AAs, i.e., 'assessment approach'). These were not presented to participants randomly, as the authorship team felt that a logical progression through the assessment process made more sense. Statements pertained to setting up the assessment (n = 3), how to measure skills (n = 4), informing the parent (n = 4), empowering the parent (n = 4), collaborating with the parent (n = 3), goal setting (n = 4), multi-professional joint working (n = 2), cultural competency (n = 1), and working with deaf-plus infants (n = 1).

## Feedback, review and round 2

Following round 1, participants received a report that summarised the 28 statements that had achieved consensus and the 41 statements that had not, showing the quantitative, group response per item. The report explained the reduction of items in the second e-Delphi round, i.e., the 28 statements that were not going to be presented in round 2 because participants had agreed they were essential and of highest importance, and therefore would form part of the proposed assessment tool. There were no eliminated or discarded statements, i.e., where 80% of participants rated a statement as a 1 ('no importance') or 2 ('low importance').

Statements that did not achieve high or low consensus in round 1 were then reviewed, with their wording scrutinised carefully to ensure that ambiguity was not a possible reason for items not reaching consensus (see data analysis and PPI sections below). Modifications to statements were made and re-presented for rating in round 2, as seen in other studies [22,30,31]. As before, optional open text boxes were offered for participants to justify answers. Fig 1 presents a flow chart illustrating the stages and outcomes of the Delphi process.

## Data analysis

There is limited guidance on the methods to analyse and present data within e-Delphi studies. For both round 1 and 2, we decided upon percentage agreement, medians, and interquartile ranges (IQR) as these descriptives are best suited for non-parametric data.

Awareness of divergence (i.e. IQR>1) and participants' qualitative feedback from round 1 was important for rewording the statements that lacked consensus for use in the subsequent survey round. For each of these statements, key words and recurring themes were highlighted and best attempts to use participants' own words for the restructuring of the statements were implemented. This work was led by the first author, but all qualitative feedback was reviewed by the authorship team and the PPI group, and new statements agreed upon in meetings.

Following round 2, the stability of statements not reaching the 80% consensus cut-off were analysed. Changes from round 1 to round 2 in the percent agreement measure, the median (measure of central tendency and group opinion) and the IQR (changes in dispersion and variation of opinion) were reviewed per statement. There were six borderline statements within 5% of the consensus margin. These statements were considered for inclusion. See 'After Round 2' in the PPI section below.

## Patient and public involvement (PPI)

A group of patient and public research partners have been involved in this NIHR funded research project (i.e., the development of an assessment tool) since May 2020. They are eight hearing and deaf professionals and nine hearing parents of deaf children. The lead author has

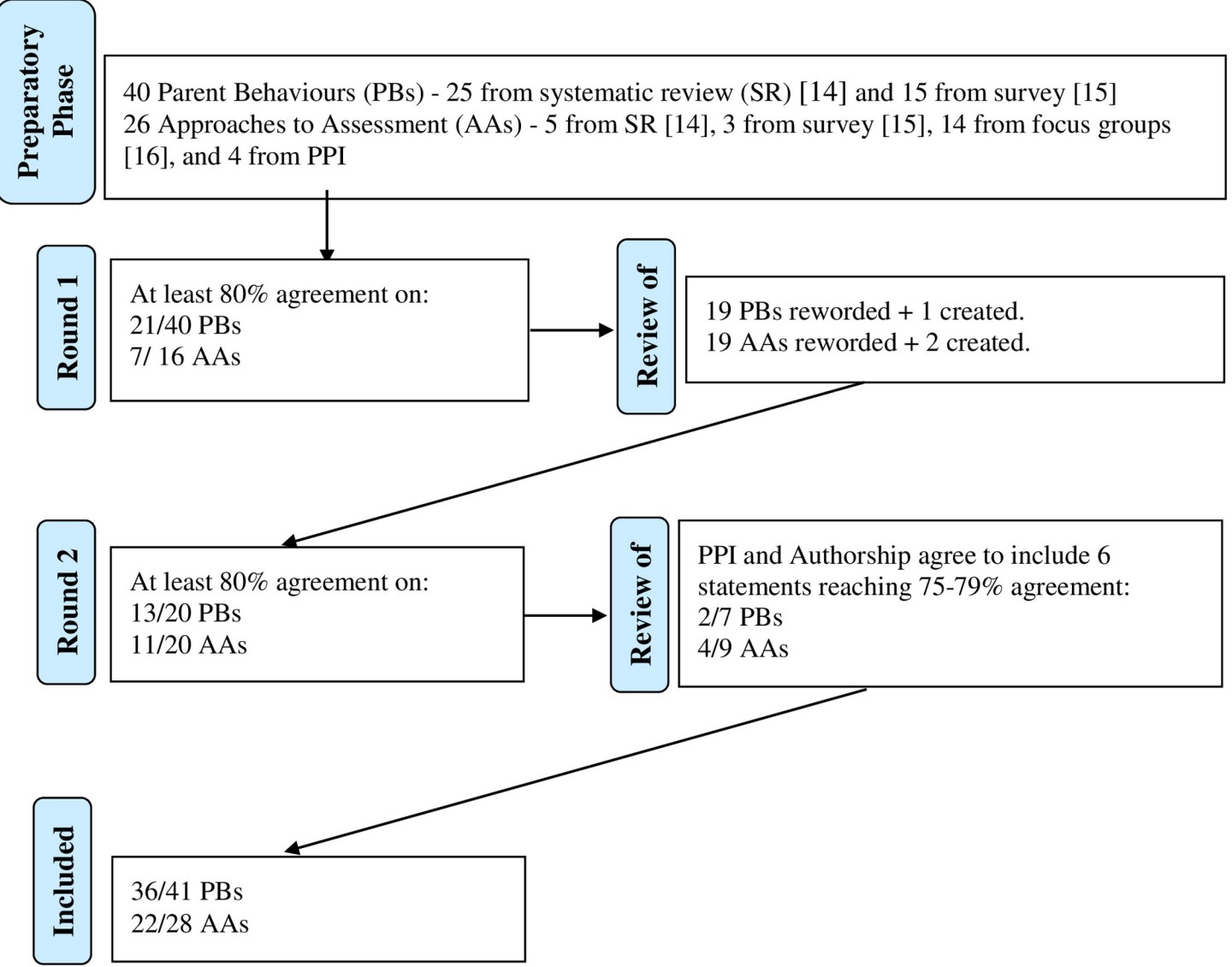

**Fig 1. Flow chart illustrating the stages of the e-Delphi process.** Note the denominator of total number of PBs and AAs changes from the 'Preparatory Phase' to the final 'Included' phase as there were some additional statements created in the 'Review of R1' phase.

met with both groups separately and as a whole team throughout the project and they have each made contributions that has helped the work become more family-centred, and relevant to deaf infants, their families and the professionals that support them. For this e-Delphi study a considerable amount of collaboration took place with the PPI groups:

**Prior to round 1: Remote review for PPI professionals.** Videos were sent in spoken English and British Sign Language (BSL) outlining the project, explaining the e-Delphi study design, and the proposed statements for round 1. Professionals were requested to either comment on a live, online shared document or send their comments via video.

**Prior to round 1: Online meeting for PPI parents.** The systematic review (14), mixed-methods study findings (15–16) to date, the e-Delphi design and proposed statements were presented. Then whole group and smaller, break-out group discussions were had relating to 1)

the wording of the current statements and 2) missing items, i.e., statements that should be added from a parent's perspective. Four statements from parents were generated for inclusion.

**After round 1: Two online meetings (one with parents and another for professionals).** In an evaluation form from a previous meeting, some parents had requested more parent-to-parent groups as the combined larger group felt very formal at times. Therefore, separate meetings were held to allow each group to speak freely, without anxiety or perceived feelings of judgement. In these meetings, statements that did not reach consensus in round 1 were reviewed. Breakout rooms were created, and one group worked on rewording the parent behaviours, and another group on assessment approaches. Reworded statements were then revisited by the authorship team before circulating round 2 to e-Delphi participants.

**After round 2: Whole group, online meeting with professionals and parents.** Following the data analysis and review of the statements that did not reach consensus after two rounds, the PPI group were presented with the six statements mentioned at the end of the data analysis section, along with the e-Delphi participants' qualitative feedback. Mostly, participants' qualitative comments spoke to small concerns with elements of the statements, which if removed, were likely to then appeal to professionals and move the statements to within the consensus threshold. Small edits or expanded explanations were decided upon for each statement by two or three members of the PPI group, using participants' feedback for each item. These are presented in the results section (see final table).

## Results

### Participants

Table 1 presents participant characteristics in terms of primary profession (many had dual roles), location, hearing status, and years of experience. Eighty-three (83) of the 95 registrants completed round 1. There was a relatively equal number of SLTs (n = 30) and academics (n = 28), with fewer QToDs (n = 22), NDCAMHS professionals (n = 2) and Psychologists (n = 1). Professionals were from thirteen different countries, with most participants contributing were from England (n = 50) and then the USA (n = 12). Most professionals were hearing (n = 78, 93%). On average, each panel member had nearly 20 years' experience of working with deaf children. Seventy-two (72) of the 83 panel members were happy for their contribution to the study (and the assessment tool's development) to be acknowledged and their names are listed in the acknowledgements section.

**Table 1. Participant characteristics.**

| Profession | 100% (n = 83) and Country where based |
|---|---|
| SLT | 36% (30)–(England = 27, Scotland = 2, Malaysia = 1) |
| Academic | 34% (28)–(USA = 11, England = 5, Germany = 3, Belgium = 2, Sweden = 2, Italy = 2, Austria = 1, Canada = 1, Netherlands = 1) |
| QTOD | 27% (22)–(England = 15, Scotland = 2, Wales = 3, USA = 1, Northern Ireland = 1) |
| Mental Health Services | 2% (2)–(England = 2) |
| Psychologist | 1% (1)–(England = 1) |
|  |  |
| **Hearing Status** |  |
| Hearing | 93% (78) |
| Deaf / Hard of Hearing | 7% (6) |
|  |  |
| **Years of Experience (Group Average)** | 19.7 (min 10 years and max 51 years) |

## Round 1

Table 2 lists the statements where consensus was reached in round 1, their levels of consensus, the category the statement is in, and each statement's median and interquartile range (IQR).

**Parent behaviours.**   In this first round, 21 of the 40 (53%) PB statements reached consensus, and percentages of agreement ranged from 80–99%, with IQRs <1. All eight statements within the parental sensitivity category gained consensus in this round, with seven of the eight statements in the top ten for highest percentages of agreement (94–99%). A parent being attuned or emotionally sensitive to the child's emotions and behaviour had the highest figure of agreement overall (99%) with a parent being genuinely interested and involved also highly rated (98%).

All four statements related to joint engagement also reached consensus in round 1. Experts agreed on the importance of balanced turn taking (98%), maintaining joint engagement (98%), on the parent waiting for the child to initiate (94%), and the parent and child face watching one another (84%).

The remaining nine statements that reached consensus were linked to how a parent can increase a child's access to language (1 of 4 statements achieved consensus) and how a parent can enrich their deaf child's language (8 out of 19 statements achieved consensus). Experts agreed that a parent expanding on their child's language (91%) and the parent interpreting the child's behaviours with language (84%) were some of the most important skills to assess in PCI. Likewise, for access to language, 83% of participants felt it was important to assess the parent's skills in physically positioning themselves at the child's level.

There were 19 parent behaviours that did not reach consensus. None of the five statements on 'attention getting' reached consensus in round 1, and many participants gave comments as to why the statements did not work in their current form. Participants shared that a parent using auditory based attention getting strategies with an unaided deaf child would not be effective, and likewise a hearing parent using visual or tactile attention getting strategies might not be the most effective way of gaining the attention of a deaf child who is a good user of their hearing equipment. In addition, there were 3 parent behaviours related to accessible language and a further 11 parent behaviours linked to language enrichment that did not reach consensus.

**Approaches in assessment.**   Seven of the 26 (27%) statements related to assessment approach reached consensus in round 1. Participants agreed all four statements linked to goal setting were important and essential. These stated that assessments should lead to goal planning (80%), goals should be jointly discussed and agreed between the parent and professional (96%), the purpose of the parent-focused goal and impact on the child should be discussed between the parent and professional (98%) and parent-focused goals should be regularly reviewed and updated (96%).

Two of the four statements linked to developing an informed parent gained consensus. These were a parent being made aware of the purpose of the assessment (96%) and the parent having the opportunity to watch back and review a video of their PCI with a professional (91%). Lastly, one of the four statements on developing an empowered parent reached consensus. This referred to providing parents with the opportunity to assess themselves with encouraging support from a professional (88%).

Consensus was not reached on the 19 assessment approaches. These were three statements on setting up the assessment (n = 3), how to measure skills (n = 4), collaborating with parents (n = 3), multi-professional joint working (n = 2), cultural diversity (n = 1), and deaf infants with additional needs (n = 1).

**Table 2. Round 1 - consensus met (i.e., rated a '4' or a '5' by ≥ 80%).**

| Parent Behaviours to Assess (n = 21) | Category | Consensus % agreed | Median (IQR) |
|---|---|---|---|
| PB17 Parent is attuned and adaptive to the child's emotions and behaviour. In research, this is called emotional sensitivity. | Parental Sensitivity | 99% | 5 (4–5) |
| PB11 Parent engages in balanced communicative turn taking (verbal or visual). | Joint Engagement | 98% | 5 (4–5) |
| PB14 Parent is genuinely interested and involved. In research this is called availability. | Parental Sensitivity | 98% | 5 (4–5) |
| PB9 Parent maintains joint engagement with their child. | Joint Engagement | 98% | 5 (4–5) |
| PB16 Parent responds to their child with on-topic behaviour or language. In research, this is called responsivity and/or contingent talk. | Parental Sensitivity | 96% | 5 (4–5) |
| PB19 Parent predominantly strives for a positive connection with their child. In research, this can be referred to as consistency or emotional regulation. | Parental Sensitivity | 96% | 5 (4–5) |
| PB13 Parent shows enthusiasm, warmth. In research this is called positive regard. | Parental Sensitivity | 95% | 5 (4–5) |
| PB18 Parent interacts with appropriate pace, play, and language for the child's age/stage. In research, this is referred to as structure and stimulation. | Parental Sensitivity | 95% | 5 (4–5) |
| PB10 Parent pauses or waits to give time for the child to initiate. | Joint Engagement | 94% | 5 (4–5) |
| PB15 Parent follows their child's lead. In research, this is called non-intrusiveness. | Parental Sensitivity | 93% | 4 (4–5) |
| PB31 Parent expands on their child's language by adding 1 or 2 more words or signs. | Language Enrichment | 91% | 4 (4–5) |
| PB38 Parent uses praise / encouragement. | S Parental Sensitivity | 91% | 5 (4–5) |
| PB26 Parent interprets their child's behaviour with language. | Language Enrichment | 89% | 4 (4–5) |
| PB27 Parent uses a range of different word types (i.e., nouns, verbs, adjectives). | Language Enrichment | 87% | 4 (4–5) |
| PB25 Parent comments on or describes an action, an object, a picture, etc. | Language Enrichment | 85% | 4 (4–5) |
| PB6 Parent and child mutually face-watching one another. | Joint Engagement | 84% | 4 (4–5) |
| PB8 Parent alerts their child to, or explains, environmental sounds (where appropriate). | Language Enrichment | 84% | 4 (4–5) |
| PB20 Parent physically positions themselves at the child's level. | Language Access | 83% | 4 (4–5) |
| PB35 Parent uses repetition (of their and/or their child's language, gesture, or vocalisations). | Language Enrichment | 82% | 4 (4–5) |
| PB36 Parent offers and labels choices. | Language Enrichment | 80% | 4 (4–5) |
| PB24 Parent labels items. | Language Enrichment | 80% | 4 (4–5) |
| **Approaches to Use in the Assessment (n = 7)** | | | |
| AA65 The purpose of the parent-focused goal, and its impact on their deaf child, should be discussed with the parent. | Goal Setting | 98% | 5 (5–5) |
| AA66 Parent-focused goals should be regularly reviewed and updated. | Goal Setting | 96% | 5 (4–5) |
| AA46 Parents should be made aware of the purpose of the observation / assessment. | Informed Parent | 96% | 5 (5–5) |
| AA63 Parent-focused goals should be jointly discussed and agreed between the parent and professional. | Goal Setting | 96% | 5 (4–5) |
| AA51 Parents should have the opportunity to watch back and review video recordings of their parent-child interaction with a professional (within the same visit if appropriate). | Informed Parent | 91% | 5 (4–5) |
| AA53 Parents should be given the opportunity to assess themselves, with encouraging support from a professional. | Empowered Parent | 88% | 4 (4–5) |
| AA62 Assessment outcomes should lead to goal setting | Goal Setting | 80% | 4 (4–5) |

**Review and rewording.** Across both sections of round 1, there were no statements that were eliminated or discarded, i.e., where 80% of participants rated a statement as a 1 ('no importance') or 2 ('low importance'). Therefore, all 19 PB statements and 19 AA statements

were reviewed by the authorship team and the PPI group. There were seven PBs and six AAs where more divergent responses were present (i.e., IQRs >1). These are marked with an asterisk in the tables below. All of the statements, but particularly those with dissenting judgements, were carefully scrutinised using participants' qualitative responses to aid the rewording of items. The review and rewording process resulted in less divergence in round 2. S1 Table displays the changes in wording for all 38 statements from their original round 1 presentation to how they were presented to participants for round 2.

Within the parent behaviours section, the main changes included creating different categories linked to *child language use* rather than parent skill. Participants had made it clear in their qualitative feedback in round 1 that some PBs were dependent on the language choice made by parents. For example, attention getting statements PB1, PB3, PB4 and PB5 were grouped together, and participants were informed that these would be most relevant where the deaf child benefitted from access to lip patterns and facial expressions. Similarly, attention getting statement PB2 and access to language statements PB21, PB22 and PB23 were grouped together, and participants were informed that these skills would be most relevant for deaf children who have access to sound and the potential to understand and use spoken language. Following the review of participants' comments, it was decided that the remaining 11 reworded statements were relevant for all deaf children, regardless of listening skills and language choices. Two prominent reframes of statements within this section were PB30 'Parent reduces questions and increases on-topic comments' and PB34 'Parent uses open questions'. Participants' feedback had the following themes: PB30 said two things at once; it is impossible for parents not to ask questions; a professional might only want a parent to reduce questions if there is overuse, not all question types negatively impact language use; it is the balance between parent and child that is important, and balance between questions and comments is also key for parents. As they are linked, feedback on PB34 was similar but participants also made the case that open questions needed to be asked when the child was developmentally ready. It was decided that PB30 would be reworded and focused on question type: 'Parent uses open questions in favour of closed questions' and that PB34 would be more about the *balance* of a parent's language use: 'Parent balances open questioning with on-topic comments'. Disclaimers about the child's developmental stages of cognition and language were added to PB28, PB29, PB32, and PB39. Lastly, one new parent behaviour was added to the 'access to language' category and this was about the parent's use of child-directed speech or sign to pique the child's interest. Therefore, for round 2, there were 20 parent behaviours to rate.

Within approaches to assessment, the categories and 'essence' of each statement mostly stayed the same, but statements were expanded to provide more clarification or better examples. Other adaptations included using softer, conditional terms such as 'where appropriate' and 'could' instead of 'should'. This was to acknowledge participants' feedback that professionals work with a highly heterogenous population and approaches in assessment would be context-bound. One statement that underwent extensive review was AA61, which related to families whose home language was not the officially recognised native language of the country in which they resided. This statement was split into three to gain consensus on whether: families should be observed in their home language (AA61a); professionals should acknowledge the culture of the family to prevent misinterpreting PCI assessment results (AA61b); and even where there may be cultural differences at play, all families could benefit from adapting their communication behaviours if unhelpful for the language development of the deaf child (AA61c). The team also added an optional question related to AA61abc with an open-text response to gain information on how participants have worked successfully in this area. From review of round 1 feedback and the preliminary work [14–16], it was decided that an additional free text box would also be added to AA50 (related to deaf-plus infants), as more expert

insight would be beneficial for working with this population. Therefore, for round 2, there were two free text boxes and 21 statements on approaches in assessment.

## Round 2

For round 2, 81 of the 83 (98%) participants returned. Table 3 lists the statements that reached consensus in round 2, the category the statement is in, the percentages of agreement, and each statement's median and interquartile range (IQR). Consensus data from round 1 is also displayed, along with the change in percentage of agreement per item across the two rounds.

**Parent behaviours.**   In the second and final round, a further 13 of the 20 (65%) reworded parent behaviour statements reached consensus, with agreement ranging from 80–95% and an IQR<1. This included six of the seven previously divergent PBs (i.e., IQR>1). Seven out of the 13 remaining statements within the language enrichment category, three out of the five remaining statements from the attention getting category, and three out of four statements remaining on access to language reached consensus. In this round, PB12 ('parent waits and watches their child's behaviours and gaze, using the child's cues to tailor the language they will use') gained the highest level of consensus (95%). This was a 23% increase from its agreement figure in round 1 and the only parent behaviour statement in round 2 that reached a consensus above 90%.

Many of the reworded parent behaviours discussed in the 'Review and Rewording' section above reached consensus. These were five of the seven statements re-grouped by child language (PB1, PB5, PB21, PB22, PB23), the new entry on child-directed language ('PBnew') and the reframed statements on questions and comments (PB30 and PB34).

Two of the largest increases in percentage agreement were for PB39 ('Parent provides language input that is appropriate to child's developmental stage') and PB7 ('Parent watches and waits when the child looks away, to allow the child to explore, to take a rest from interacting, to take a turn in initiating'). PB39 moved from 31% agreement (round 1) to 88% agreement (round 2), an increase of 57%, two points on the scale in terms of median score and had less divergence (IQR = 2 to IQR = 1). PB7 gained 28% of participants' agreement (round 1) but achieved 83% agreement (round 2). The smallest change observed between round 1 and round 2, was PB28 ('Where appropriate, parent uses mental state verbs i.e., 'like', 'know', 'think'). In round 1, the statement achieved 77% (median 4, IQR 4–5) and in round 2, the statement achieved 81% (median 4, IQR 4–5). Due to the slight decrease in the number of expert participants between round 1 and round 2, this is likely to mean that one more person rated the statement a 4 or 5.

There were seven out of 20 (35%) reworded parent behaviour statements that did not achieve consensus. Table 4 displays these ratings and the observable changes from round 1 to round 2.

Five of these statements had an increase in the percentage of agreement between participants, with changes in agreement ranging from 6 to 29%. Two statements decreased in their agreement percentages, i.e., their agreement level lowered following rewording. These were PB4 (parent uses multiple strategies at one time to gain a child's attention) and PB40 (parent uses touch as a tactile way of highlighting speech, tone, or rhythm in language). For PB4 (divergent in round 1 but not in round 2), participants argued that one mode at a time would be enough–auditory strategies, visual strategies, or tactile strategies. This would allow a parent to monitor which approach in attention-getting the child responds to best, so that the approach is child-led, and the child is not overwhelmed or over-stimulated by multiple methods. For PB40, many participants thought this was quite unnatural and intrusive for PCI and play, but rather a helpful strategy in word learning and/or syllable counting. Participants also

**Table 3. Round 2 –reworded statements achieving consensus (n = 24).**

| Parent Behaviours to Assess | Round 2 rating | Round 2 Median (IQR) | Round 1 rating | Round 1 Median (IQR) | Category | ⇅ |
|---|---|---|---|---|---|---|
| PB12 Within the interaction, parent waits and watches their child's behaviours and gaze, using the child's cues to tailor the language they will use. | 95% | 5 (4–5) | 72% | 4 (3–5)* | Language Enrichment | ↑ 23% |
| PB5 Where the child benefits from access to lip patterns, facial expressions, and/or visual perception of sound, parent actively waits or pauses their communication until their child looks at them. | 89% | 4 (4–5) | 69% | 4 (3–5)* | Attention Getting | ↑ 20% |
| PB new Parent uses appropriate child-directed language (e.g., exaggerated, or tuneful intonation, exaggerated facial expressions, gesture, a larger signing space). | 89% | 4 (4–5) | N/A | N/A | Language Access | N/A |
| PB34 Parent balances open questioning with on-topic comments. | 88% | 4 (4–5) | 68% | 4 (3–5)* | Language Enrichment | ↑ 20% |
| PB39 Parent provides language input (i.e., average number of signs/ words) that is appropriate to child's developmental stage. | 88% | 5 (4–5) | 31% | 3 (2–4)* | Language Enrichment | ↑ 57% |
| PB1 Parent uses visual attention-getting strategies (e.g., moving into the child's visual field). | 88% | 4 (4–5) | 68% | 4 (3–4) | Attention Getting | ↑ 20% |
| PB21 Where the child is using/developing skills in spoken language, parent uses appropriate voice volume. | 86% | 4 (4–5) | 76% | 4 (4–5) | Language Access | ↑ 10% |
| PB7 Parent watches and waits when the child looks away, to allow the child to explore, to take a rest from interacting, to take a turn in initiating. | 83% | 4 (4–5) | 28% | 3 (2–4)* | Language Enrichment | ↑ 55% |
| PB2 Parent uses auditory attention-getting strategies (e.g., using the child's name). | 82% | 4 (4–5) | 72% | 4 (3–4) | Attention Getting | ↑ 10% |
| PB23 In earlier stages of development, where the deaf child has access to spoken language, parent makes accompanying sounds to the child's action/ toys/ items. | 82% | 4 (4–5) | 66% | 4 (3–4) | Language Enrichment | ↑ 16% |
| PB28 Where contextually and pragmatically appropriate (developmental stage / relevant moment), parent uses mental state verbs (i.e., 'like', 'know', 'think') within the interaction. | 81% | 4 (4–5) | 77% | 4 (4–5) | Language Enrichment | ↑ 4% |
| PB30 Parent uses open questions in favour of closed questions. | 80% | 4 (4–4) | 76% | 4 (4–5) | Lang Enrichment | ↑ 4% |
| PB22 Where the child is using/developing spoken language, parent is mostly within 1 to 2 metres of the child's amplification device(s) where possible. | 80% | 4 (4–5) | 70% | 5 (3–5)* | Language Access | ↑ 10% |
| **Approaches to Use in the Assessment** | | | | | | |
| AA60 Where a family has more than one professional involved, the assessing professional should share information from the assessment with the rest of the team. This will reduce duplication of assessment and allow quicker access to intervention / support. | 95% | 5 (4–5) | 58% | 4 (3–4) | Joint Work | ↑ 37% |
| AA61b The culture of the family should be acknowledged when observing parent-child interaction, to prevent the professional misinterpreting assessment results. | 94% | 5 (5–5) | AA61 65% | 4 (3–4) | Cultural Diversity | ↑ 29% |
| AA61a Families should have their parent-child interaction observed in the language of the home, with assessors using interpreters or bilingual co-workers to understand the language used. | 93% | 5 (5–5) | AA61 65% | 4 (3–4) | Cultural Diversity | ↑ 28% |
| AA50 For some children, e.g., those with additional or complex medical needs, the activities within parent-child interaction assessments may need to be more flexible and varied, i.e., whenever the child is most interactive within their daily routines. | 90% | 5 (4–5) | 63% | 4 (3–4) | Deaf-Plus | ↑ 27% |
| AA52 The review of the parent-child interaction assessment should be largely strength-based, i.e., identifying what is working well. There could also be scope to sensitively highlight behaviours with potential to improve, as long as the overall review is positive and encouraging. | 89% | 4 (4–5) | 77% | 4 (4–5) | Empowered Parent | ↑ 12% |
| AA64 Goals should be mostly focused on a parent's current strengths in the assessment. The parent may also wish to pick an important behaviour they would like to practice / become more confident with. | 87% | 4 (4–5) | 77% | 4 (4–5) | Empowered Parent | ↑ 10% |
| AA61c All parents, even where there may be cultural differences at play, may benefit from adapting their communication behaviours if unhelpful for the language development of the deaf child. The review of an assessment video can assist with these discussions. | 86% | 4 (3–4) | AA61 66% | 4 (3–4) | Cultural Diversity | ↑ 20% |

*(Continued)*

**Table 3.** (Continued)

| Parent Behaviours to Assess | Round 2 rating | Round 2 Median (IQR) | Round 1 rating | Round 1 Median (IQR) | Category | ↑↓ |
|---|---|---|---|---|---|---|
| AA43 To accurately capture and then reflect on parent-child interaction, a video recording is recommended at least once in parent/professional partnership work. Timing of when this formal measure is taken will depend on parental well-being, parental personality and the strength and trust within the parent/professional relationship. | 84% | 4 (4–5) | 69% | 4 (3–5)* | Informed Parent | ↑ **15%** |
| AA45 Parents could be encouraged to send videos to an early intervention provider for review, where the professional is not present (especially if the child has additional needs, the child does not engage, or parents require support within a particular context). | 80% | 4 (4–4) | 64% | 4 (3–4) | Empowered Parent | ↑ **16%** |
| AA56 Where possible, the parent should be offered the choice of receiving a copy of the parent-child interaction recording, following the assessment session with the professional. | 80% | 5 (4–5) | 57% | 4 (3–5)* | Informed Parent | ↑ **23%** |
| AA48 Where possible, parents should be asked where they would prefer to be observed. | 80% | 4 (4–5) | 74% | 4 (3–5)* | Collaborate | ↑ **6%** |

*An asterisk signifies divergent views within the response, i.e., an IQR >1.

felt that this strategy may be more appropriate for a therapist or teacher to use in direct work, because parents should not feel they need to be teachers during interactions with their infants.

A closer look at the low agreement statements highlighted two parent behaviour statements within the language enrichment category which were very near to achieving consensus: PB29 (parent talks ahead of actions or events; 77%), and PB32 (parent rephrases a child's incorrect grammar; 76%). Both statements had increased in their percentage agreement figures (PB29 by 12% and PB32 by 7%) and in their spread of opinion when compared to round 1 results (IQRs of 3–4 in round 1 and IQRs of 4–5 in round 2). Qualitative feedback for PB29 was either that is not always possible to pre-warn a child of a change and that this parent behaviour is dependent on the age and stage of the child. For PB32, participants were keen to note that PCI should not include corrective grammar lessons as this can impact bonding and self-esteem. The remaining five parent behaviour statements that did not achieve consensus (PB4, PB3, PB33, PB37, and PB40) were considerably lower in percentage agreement (ranging from 62–13%) with a substantial gap from the statements bordering close to consensus, suggesting clear views from experts that the remaining five were less essential and less important items. Further, PB37 became divergent in round 2, i.e., an IQR>1. Through discussion, across the authorship team and in PPI meetings, it was agreed that statements PB29 and PB32 would be retained in the final list of statements and reworded to include caveats noted by participants in their qualitative feedback (see Table 5). Conversely the other five parent behaviours were not core and not included.

**Approaches in assessment.** Eleven of the 21 (52%) reworded statements related to assessment approach reached consensus in round 2. This included three of the six previously divergent AAs (i.e., IQR>1). These were all three statements on cultural diversity (AA61a,b,c), the statement on deaf children with additional needs (AA50), the remaining two statements on developing an informed parent (AA43, AA56), the remaining three statements on empowering the parent (AA52, AA64, AA45), one of the three statements on collaborating with parents (AA48), and one of the two statements on multi-professional joint working (AA60).

The four statements with the highest percentages of agreement were also the four statements with the highest changes in agreement scores between round 1 and round 2. AA60 on joint working (professionals sharing information) had the highest number of agreement (95%) with a 37% increase from round 1 (58%). In their comments, participants agreed that with

**Table 4. No consensus reached following rounds 1 and 2 (n = 17 statements).**

| Parent Behaviours to Assess | Round 2 rating | Round 2 Median (IQR) | Round 1 rating | Round 1 Median (IQR) | Category | ⇅ |
|---|---|---|---|---|---|---|
| PB29 Parent informs the child of an action or event ahead of doing it, using a range of visual cues if appropriate for the child's understanding. | 77% | 4 (4–5) | 65% | 4 (3–4) | Language Enrichment | ↑ 12% |
| PB32 Within the interaction, parent supportively rephrases deaf child's language with correct grammar (where contextually and pragmatically appropriate, i.e., developmental stage, a natural moment). | 76% | 4 (4–5) | 69% | 4 (3–4) | Language Enrichment | ↑ 7% |
| PB4 Parent uses multiple strategies at one time to gain the child's attention (e.g., moving into the child's visual field and saying 'wow', tapping and saying the child's name). | 62% | 4 (3–4) | 74% | 4 (3–5)* | Attention Getting | ↓ 12% |
| PB3 Parent uses tactile attention-getting strategies (e.g., tapping). | 51% | 4 (3–4) | 45% | 4 (3–4) | Attention Getting | ↑ 6% |
| PB33 Parent rephrases their child's language into a question, i.e., the child says/signs "cake" and the parent rephrases into "Can I have cake daddy?' | 47% | 3 (3–4) | 18% | 3 (2–3) | Language Enrichment | ↑ 29% |
| PB37 Parent models mistakes in their own language if/when they arise, i.e., 'The fireman is crying… I mean climbing! I used the wrong word/sign'. | 33% | 3 (2–4)* | 22% | 2 (2–3) | Language Enrichment | ↑ 11% |
| PB40 Parent uses touch as a tactile way of highlighting speech / tone / rhythm in their language (e.g., parent says 'Hel-lo Ma-ya' with taps for each syllable). | 13% | 2 (2–3) | 40% | 3 (3–4) | Language Access | ↓ 27% |
| **Approaches to Use in the Assessment** | | | | | | |
| AA49 As well as observing interaction in play, professionals could sample interactions within daily routines (e.g., mealtimes, dressing) where parents are willing. | 78% | 4 (4–5) | 78% | 4 (4–5) | Set up | - |
| AA47 Where possible, assessments of parent-child interaction should take place in the child and parents' most natural, most familiar settings. | 76% | 4 (4–5) | 69% | 4 (3–5)* | Set up | ↑ 7% |
| AA44 Though a video recording of 10 minutes of interaction should provide enough material for watch back and reflection, the length of a video recording should be discussed with parents as they may request more or less time. | 75% | 4 (4–5) | 50% | 4 (3–4) | Collaborate | ↑ 25% |
| AA57 If the family requests or the context deems it necessary, all main caregivers (i.e., mothers, fathers, grandparents, older siblings) should be given the opportunity to have their interaction skills observed and reflected upon. | 75% | 4 (4–4) | 66% | 4 (3–5)* | Collaborate | ↑ 9% |
| AA42 Joint engagement could be observed by noting how long a parent and child remain connected. In some cases, it may be appropriate to estimate this, particularly for the purpose of reviewing progress. | 66% | 4 (3–4) | 40% | 3 (3–4) | Measuring skills | ↑ 26% |
| AA41 Evaluating joint engagement could be observing the connected turns between parent and child. In some cases, it may be appropriate to count these turns. | 65% | 4 (3–4) | 58% | 4 (3–4) | Measuring skills | ↑ 7% |
| AA54 Parents and professionals could reflect on each parent behaviour together using scales. Professionals could describe each parent behaviour before the parent reflects on their interactions. The wording of the scale to be parent-centred and positively framed. | 63% | 4 (3–5)* | 58% | 4 (3–4) | Measuring skills | ↑ 5% |
| AA58 To reflect everyday language in the home, the observation may need to take account of, and potentially include, other siblings present at home with the deaf child. | 64% | 4 (3–4) | 57% | 4 (3–4) | Set up | ↑ 7% |
| AA55 Parent-child interaction behaviours can be presented as a list with the parent and professional discussing and then selecting which ones they use and feel confident with. | 57% | 4 (3–4) | 32% | 3 (2–4)* | Measuring skills | ↑ 25% |
| AA59 Where appropriate, deaf professionals (e.g., deaf teachers of deaf children, deaf language specialists, deaf professionals working in mental health services) are recommended to be involved in the assessment of parent-child interaction where possible. | 55% | 4 (3–5)* | 38% | 3 (3–4) | Joint Work | ↑ 17% |

An asterisk signifies divergent views within the response, i.e., an IQR >1.

consent, information sharing is essential for reducing stress for families. The next statements with high levels on consensus were two items on cultural diversity. First AA61b (acknowledging the culture of the family) achieved 94% consensus and AA61a (observing PCI in the home

**Table 5. Reworded statements following the second and final e-Delphi round (n = 6).**

| Parent Behaviours: Round 2 Statements | Reworded and Included Following PPI Review |
|---|---|
| PB29 Parent informs the child of an action or event ahead of doing it, using a range of visual cues if appropriate for the child's understanding. | PB29 **Where necessary or possible**, parent informs the child of **next steps or a change** using a range of visual cues appropriate for the child's understanding, **i.e., parent leaving the room for water.** |
| PB32 Within the interaction, parent supportively rephrases the deaf child's language with correct grammar (where contextually and pragmatically appropriate, i.e., developmental stage, a natural moment). | PB32 Parent **models** the correct grammar back to a child for what they have just said. For example, the child says, 'Teddy eat' and the parent would say 'Yes, Teddy is eating'. **There is no expectation the child will repeat back what the parent has said. The rephrase should be developmentally appropriate and parents should avoid overuse of this behaviour.** |
| AA49 As well as observing interaction in play, professionals could sample interactions within daily routines (e.g., mealtimes, dressing) where parents are willing. | AA49 **Giving parents the opportunity**, professionals could **observe** interactions within daily routines (e.g., mealtimes, dressing) as well as observing interaction in play. |
| AA47 Where possible, assessments of parent-child interaction should take place in the child and parents' most natural, most familiar settings. | AA47 Where possible, observing parent-child interaction should take place in **the families' chosen optimal setting, where the child will be most communicative**. |
| AA44 Though a video recording of 10 minutes of interaction should provide enough material for watch back and reflection, the length of a video recording should be discussed with parents as they may request more or less time. | AA44 A video recording of **up to 10 minutes** of interaction should provide enough material for watch back and reflection. The length of a video recording should be discussed with parents as they may request more or less time. |
| AA57 If the family requests or the context deems it necessary, all main caregivers (i.e., mothers, fathers, grandparents, older siblings) should be given the opportunity to have their interaction skills observed and reflected upon. | AA57 If the family requests or the context deems it necessary, all main caregivers (i.e., mothers, fathers, grandparents, older siblings) **could** be given the opportunity to have their interactions observed and reflected upon. |

Bold text signifies altered text after round 2 to incorporate participant comments/ reflections/ concerns/ caveats.

language) achieved 93%, their changes in percentage agreement were 29% and 28% respectively. For these items, participants suggested that it would be best to involve professionals that share the culture of the family as they can inform the assessment, and make families feel more confident and comfortable using their own language. This would increase the validity of the observation. Some participants suggested that the PCI assessment could be done in the home language, and then if the parent speaks English, they could translate for the professional upon watching back. Another important point for professionals unable to work with bilingual co-workers or cultural brokers was that whilst it is important professionals understand the culture of the family, they should not make any general assumptions as variations exist. All participants' responses to the open-text question on cultural diversity are provided in S2 File.

Another statement that achieved high levels of consensus was AA50 (flexible observations when working with deaf-plus children and their families), achieving 90% agreement which was an increase of 27% and one scale point from the results for this statement in round 1 (63%). Participants commented that parents could send their own videos to the professional as this would mean there was less pressure for the parent and child to 'perform' on the visit, or that the professional could observe highly interactive moments as well as moments where communication was more limited, where parents may need more support and guidance. Participants' responses for the open-text question on working with the deaf-plus population are provided in S1 File.

There were 10 out of 21 (48%) reworded statements on assessment approach that did not gain consensus. These are displayed in Table 4. These were all three statements on setting up

the assessment, all four statements on how to measure skills two statements on collaborating with parents, and one related to multi-professional joint working. For all of these statements, more participants increased their rating between round 1 and round 2, with changes in percentage of agreement figures ranging between 5 to 26%.

As with parent behaviours, statements that did not reach consensus within assessment approaches were reviewed. There were four statements that were very close to the consensus threshold of 80%, The first was AA49 (observing daily routines) with 78%. The percent agreed, median and IQRs for this statement remained high and stable between round 1 and 2. We did not gain much qualitative feedback from participants who rated this statement as a 1, 2, or 3 on the scale, other than this was harder to achieve for professionals working in clinical settings. Our PPI group of hearing parents felt this was an essential statement to include as so much communication centres around daily routines such as mealtime, feeding, bath time, nappy changing, and dressing, therefore a slight reword was made and this statement retained.

Statement AA47 (assessing in natural, familiar settings) had a percentage agreement of 76%. Statements AA44 (video recording length) and AA57 (giving all primary caregivers the opportunity to have PCI assessed) both achieved 75%. For all three statements, the percentage agreement increase was between 7–25% and all IQRs were narrower/less divergent in round 2 results. These statements were viewed as essential by both the authorship team and the PPI group and were reworded (see Table 5). Participant feedback was used to modify statement AA47 in order to give a parent more power, choice, and convenience when deciding on optimal settings for interaction. Similarly for AA57, participants shared this would be great but unrealistic with busy caseloads and hence 'should' was replaced with 'could'. For statement AA44, many participants felt that three to five minutes was adequate and that 10 minutes of video recording was excessive. When these comments were presented to our PPI panel, parents of deaf children felt strongly that 'up to 10 minutes' should be the wording as they felt that a parent might need more than three to five minutes 'to warm up' and ignore the camera/ the professional but agreed that more than 10 minutes of video recording would be excessive. Most importantly, parents felt video recording length should be discussed and agreed upon before the filming had begun. The remaining statements on approaches to assessment (AA42, AA41, AA54, AA58, AA55, AA59) were considerably lower in percentage agreement (ranging from 66–55%). AA54 and AA59 increased in divergence. Whilst they were not included in the final list of statements, it was noted that the majority of participants (i.e., over 50%) agreed they were important. Four of these statements were about measuring PCI, one about including siblings, and another about working with deaf adults.

## Combined results

After rounds 1 and 2, there were 52 statements out of 69 (75%) that reached consensus and 16 statements (25%) that did not. As mentioned, following extensive review and discussion, six statements from the close-to-consensus group were reworded and retained (see Table 5). Therefore, a final total of 58 of the 69 (84%) statements were included for the proposed assessment tool. S2 Table displays the number of statements per category that achieved consensus in round 1, round 2 and following the post e-Delphi data analysis and review discussed above.

## Discussion

This study gained expert opinion on the core content and principles of a new assessment for PCI where the child is deaf and aged 0–3 years. Statements reviewed in the e-Delphi were based on a systematic review [14] and a studies of professional practice [15,16]. In addition, co-production work with a PPI team (17 hearing and deaf parents and professionals) was

embedded throughout each study phase. Eighty-three experts (SLTs, QToDs, NDCAMHS professionals, Psychologists and academics) working internationally with deaf infants and their families agreed on the importance of 52 statements out of a possible 69 through two rounds of voting and feedback. A further six statements were included following data analysis and discussion between the authorship team and PPI group. Experts joined from all four nations of the United Kingdom, the USA and Canada, Malaysia and six European countries. On average, each expert participant had 20 years of experience in the field. Across the two rounds, there was excellent retention of expert participants, suggesting a good estimation of the degree of consensus in the final results (i.e., minority opinions did not leave the study).

The first objective of the study was to reach consensus on which parent behaviours to include in an assessment. All statements related to *parental sensitivity* achieved high levels of consensus. The importance of parental sensitivity in PCI with deaf infants is well evidenced [2,32,33]. In their large-scale, longitudinal study of 285 deaf children with cochlear implants, Cruz and team [34] found deaf children had 1.52 years less of a language delay when their parents had above average skills in maternal sensitivity and language stimulation. Pressman and colleagues [35] found that maternal sensitivity was not correlated with initial child language scores, but correlations were present in the follow up assessments 12 months later. They calculated that maternal sensitivity predicted expressive language and had a larger positive effect on the sample of deaf children compared to their hearing sample (ibid). It is appropriate therefore, that the top five behaviours with the highest agreement were related to the parent being attuned and adaptive to the child's emotions and behaviour, the parent having a genuine interest in their child, and the parent responding to the child with on-topic behaviour or language.

All four statements on *joint engagement* also reached consensus, with balanced communicative turn taking, and maintaining joint engagement nearing total agreement. Many studies have shown the positive relationship between deaf children's language scores and time spent in co-ordinated (or mutual) joint engagement between parent and child [2,36,37], i.e., the longer a parent and child are engaged, the better the child's language will be over time. A particular behaviour linked to more successful instances of joint engagement was non-intrusiveness, i.e., following the child's interest rather than directing them [2,37]. Parental sensitivity and initiating and maintaining joint engagement are therefore intertwined and reliant one another: a parent who is interested and attuned to their child's behaviour, who shares the same interest as their child, will offer contingent comments and behaviours, and the dyad will be mutually engaged. The parent must also be aware of the balance in engagement, the space for the child to 'take a turn', verbally or non-verbally, in order to maintain the mutual participation of both child and parent. The longer the connection, the more bonding, enjoyment, and opportunities for language there can be. The fact that all sensitivity and joint engagement behaviours gained the highest levels of consensus fits with the existing literature; joint engagement and parental sensitivity are important to observe within PCI assessment.

Whilst most parent behaviours were viewed by experts as beneficial for developing spoken and/or sign languages, there were some that diverged across participants because of language choices made by parents. For example, a parent's volume of voice or proximity to the child's listening device would be more relevant for a deaf child who relies on their listening devices and is responding to and developing spoken language, than for a child who is developing sign language. Similarly, methods used to gain the child's attention may differ. For children who are responding to and developing sign language, and/or benefiting from lip patterns and facial expressions, a visual way of gaining attention may be more effective. Parents of deaf children make more frequent use of visual and tactile strategies to gain their child's attention when compared to hearing parents of hearing children [38]. In this current e-Delphi, tactile

strategies for gaining attention or using multiple modes of attention getting at one time did not reach consensus. Participants chose PBs that were child-led, to see which methods (visual or auditory) the child responded to first. Rather than tactile strategies, the more passive attempts at getting a deaf child's attention [39–43] were favoured by the e-Delphi's expert panel. These include behaviours such as the parent actively waiting or pausing their communication until the child looks. A parent who displays this behaviour may well be following the more 'non-intrusive' patterns of gaining and maintaining attention mentioned above, where the parent follows the child's lead and pace.

Many of the statements that achieved high levels of agreement were centred on the parent's response to their deaf child, rather than the parent instigating anything new themselves. Expert participants seemed to value the statements that required a parent to step back, observe, receive the child's action or utterance, and then respond. Statements achieving high agreement were the ones that promoted balance and a respect for the child's place within the interaction, a respect for their gaze, actions, interests, and contributions. This responsive, child-centred ethos in interacting with deaf infants aged 0–3 years may also explain why participants did not highly rate behaviours such as PB33 (rephrasing grammar) or PB37 (modelling or highlighting mistakes in parent's language). As many participants noted, the teaching of grammar, vocabulary, or speech production is not the job of a parent of a deaf child, particularly not at this age, and therefore it is not a vital behaviour to assess or promote in PCI work. *Modelling* language and *exposure* to grammar is important, hence our inclusion of PB33 in our final list of parent behaviours, however, parents should not be encouraging their deaf child to repeat, correct, or rephrase their utterances.

The second objective of the study was to develop best practice recommendations on *how* to conduct an assessment of PCI. Participants agreed that parents should be aware they are being assessed and have the power to choose where to be observed and what type of interaction they would like to have assessed (e.g., play, dressing, mealtime). Partnership working with parents, where the parent is seen as the expert of their child, forms the general principle of many policies and codes of practice in the UK such as the SEN Code of Practice [44] and The Best Start to Life [45]. Helpful, internationally accepted, guidance on working with families and deaf infants can be found within the position statement 'Principles and Guidelines for Early Hearing Detection and Intervention Programs' [10] and Moeller and team's paper 'Best practices in family-centered early intervention (FCEI) for children who are deaf or hard of hearing. . .' [13]. Both of these documents also highlight the importance of parent/professional partnerships and recommend the role of the professional is to be a 'supporter, partner, and coach' ([10], p.25). The current study's findings align with this philosophy; many participants agreed that PCI should be video recorded (at least once) and that parents should be given the opportunity to watch the interaction and assess themselves before receiving feedback from the supporting professional. Further, participants agreed that video feedback should be strength-based, i.e., parent and professionals identifying what is working well, but there could also be scope to sensitively highlight behaviours with potential to improve, as long as the overall review is positive and encouraging. The JCIH statement [10] recommends professionals use evidence-based practices and build on families' strengths, fostering their confidence and competence in providing a range of jointly attended-to language opportunities with regular conversational turns throughout natural daily interactions.

A goal of FCEI [13] is also the development of trusting, respectful family-provider partnerships, characterised by honesty, shared tasks, and open communication. They also recommend a focus on facilitative family–child interactions, rather than child-directed therapies. These recommendations align with our participants' agreement on collaborative conversations about goal setting, and sharing responsibility for the child's developmental outcomes. The mention

of 'family-child' interactions rather than one parent also speaks to our high agreement statements on giving all caregivers the opportunity to have their PCI assessed where appropriate and possible. For the first time, this consensus paper and the upcoming PCI assessment tool provides specific guidance on how to achieve an asset-based observation of parent-deaf child interaction, jointly reviewed by parent and professional.

Participants agreed that, for families who have a deaf-plus child, i.e., a deaf child with additional needs, more flexibility may be required for the assessment. For example, parents may find it is better suited to send the professional a video of interaction rather than video record a live visit, as the child may be more interactive without the external visitor. Likewise, participants agreed assessing less familiar daily routines should be considered and professionals should be prepared to watch less successful instances of communication (if that is the parent's choice) in order to provide assistance with daily routines. Around 40% of deaf children are reported to have an additional disability [46] which can affect each family in different ways. As per expert comments in S1 File, it is therefore important to follow each family's lead in how to assess PCI and to respect the parent as the expert of their child and their child's conditions. Including deaf-plus children in studies that record every day routines is emerging [47–49], but these studies used audio-only recording software (LENA) which limits an analysis of PCI.

Experts concurred that families should be assessed in their home language with bilingual co-workers and/or interpreters helping to interpret the findings. Emerging research is now providing evidence that deaf children can learn two spoken languages [50,51] and that professionals should not discourage bilingualism to parents [52] providing the deaf child has good access to sound. Experts also agreed that cultural diversity and family context should be acknowledged when assessing and drawing conclusions from the PCI assessment, and that if there are behaviours that are unhelpful for the development of the deaf child, these should be raised and discussed sensitively using the video recording of PCI. Government data in England [53] have shown that deaf students who have English as an additional language (EAL) do less well at school than their deaf and hearing EAL peers and the need for better family support and education has been acknowledged. As many of our expert participants shared (see S2 File), a way of connecting with and accurately assessing and supporting parents who have EAL is through the use of video, bilingual co-workers, and using culturally based songs, toys, and games in the assessment.

In this study, expert participants agreed that parents should be given the opportunity to assess their own skills via a video review, but how this is done practically did not reach consensus. The two methods presented in our e-Delphi were the use of scales or a checklist. When using a scale, the professional might describe each parent behaviour before the parent reflects on their skill competency in that area. Scales were used in 34% of the research papers within the systematic review on assessing PCI with deaf infants aged 0–3 [14] and may be more sensitive to skill change than a present/not present checklist. When using a checklist, the professional may again describe the behaviour before the parent and professional discuss which skills were observed on the video. Checklists do not feature in PCI research in deafness but are used in therapies such as Palin PCI Therapy for Stammering [54]. A checklist has the potential to be less threatening than a scale as parents are in charge, simply selecting which behaviours they observed rather than reflecting on their competence in key skills. In the current e-Delphi, the use of scales achieved a slightly higher consensus (63%) than checklists (57%), but both failed to achieve the consensus threshold. When advising on Delphi studies, Hasson and colleagues state that consensus does not necessarily mean the correct answer has been found [55] and that results can help to structure discussion, raise items for debate, and help to streamline focus. Further discussion around professionals' reservations with scales and checklists will help to better understand the responses and potential differences between professional groups.

With regards to the next stages of the PCI tool's development, how to measure parents' skills will be decided upon in co-production stages with parents of deaf children and hearing and deaf professionals working in the field.

Many participants commented that the assessment tool was a worthwhile way of closing the gap between research and practice, but it was also the case that money, time, resources, and a heterogenous population [56] often led to huge variations in services being offered to families. A new PCI assessment might be more effective in allocating resources towards each family. However, we also acknowledge that organisational and contextual factors influence practice.

Professionals had mixed views about the inclusion of others in PCI assessments, specifically siblings and deaf professionals. Professionals who argued for sibling inclusion noted these points: increased ecological validity; highly informative; siblings are essential to modelling language; sibling communication is important for the deaf child; and regular individual special time may not be possible. Professionals who rated '1', '2' or '3' provided no justification for their response, so limited insight can be gained here. We expected that if a parent has a deaf child at home along with another infant (perhaps under 5 and not in childcare), the observation might need to include siblings in order to reflect everyday language, everyday dynamics, and the natural environment. Despite its conditional wording i.e., 'the observation *may* need to include siblings', this statement was deemed less important and not essential to include in a PCI assessment.

There are three possible explanations for the mixed views around inclusion of deaf professionals. Firstly, the vast majority of panel participants were hearing. Secondly, many participants noted it was impossible to organise deaf professional inclusion because not all early intervention services employ deaf ToDs, deaf sign language instructors, deaf Language Specialists, deaf Family Support Workers or deaf mentors. Lastly, most participants were based in England, where there are no laws about the inclusion of deaf adults in early intervention. In the USA, including deaf adults in practice is federal law. If a larger sample had been recruited from that country, the statement may have achieved consensus. Both FCEI [13] and JCIH [10] recommend the inclusion of deaf adults in early intervention programs, not only for sign language instruction, but to offer families and deaf children support, guidance, and mentorship. There are multiple reported for families when deaf adults are involved in family-centred early intervention, such as reductions in stress and increased confidence [57], the opportunity to see possible successes for their children [58], learn a range of visual strategies to assist with language learning [59], and deaf adults being role models for families and deaf children [60]. If, once developed, the proposed tool is used internationally, discrepancies between countries on issues such as involving deaf adults should be included in the tool's manual.

## Limitations

The current study's findings may be biased towards the UK as 70% of participants came from this region and it has been suggested that a diverse panel leads to better outcomes as it allows for a wider range of alternatives and perspectives [61]. Whilst there are merits in the highly experienced, international, and multi-professional expert participants, we did not collect information on the ethnicities or cultural backgrounds of those recruited. Despite not knowing the cultural or linguistic backgrounds of participants, nearly all participants were recruited from western, educated, industrialised, rich, and democratic countries (WEIRD). This likely means that their research, practice and/or experiences will be grounded in a mainstream view of language acquisition for deaf infants, summarised in a systematic review [14]. This may also explain low instances of divergence. Some caution therefore must be taken when interpreting these findings more globally.

In addition, 'professionals as experts' was the focus of this study, however parents are recognised as experts in their children's lives, and their views were not reflected here. Parents' views will feature strongly in the next phase of this larger research project, through co-production.

It is important to note the inherent limitations within Delphi studies, that are also relevant to this research project. Firstly, how survey items are designed, such as their use of abstract language and sentence length, has a proven influence on Delphi outcomes [62]. The piloting phase with a range of professional groups within our study addressed this issue to some degree. Secondly, participants were self-selecting individuals utilising their opinion to answer survey items. Individual agency, values, experiences, interpretations, agendas, and social and political climates underscore the answers given within Delphis [63]. We invite the reader to hold these limitations in mind when interpreting the results.

A limitation specific to this study was not presenting the qualitative comment summaries to participants between round 1 and round 2. Providing access to this data would have provided participants with more insight into people's thinking, which may have led to more people adapting their response after seeing new, anonymous perspectives from others. Due to the number of statements and the range of feedback received, this would have meant sending participants a lengthy document to read in between rounds. We were concerned that this would create burden for those involved and may have led to attrition. Whilst the format or length of feedback has not been investigated, it is known that a higher number of items in e-Delphis is significantly associated with reduced response in the second round [64]. We therefore opted for a six-page feedback document with a short summary page at the beginning, followed by all statements and their percentage agreement.

Finally, the a-priori consensus threshold may have been better decided upon a-posteriori, or perhaps a flexible range suggested instead, e.g., items above 80% would be considered as 'consensus achieved' and items between 75–79% would be reviewed by the authors and involvement group. The decision to choose a higher cut off for consensus was influenced by the overview of systematic reviews from Niederberger and Spranger [23] where they highlighted that lower consensus thresholds mean more participants do not agree with the consensus, risking the neglect of relevant and unusual judgements. We wanted consensus to be based on high levels of agreement between experts, but also reflect the views of our parent PPI group. In their systematic review of 30 Delphi studies, Junger and team [22] found that percentage of agreement was usually 75 or 80% but that some studies used the stability of group response over successive rounds or a cut off inclusion based on a 'natural break' in the overall scores. With hindsight, a cut off based on a natural break may have been a better choice than an a-priori figure as there was a noticeable gap between our borderline statements 75–79% and the other statements with low agreement. For the purposes of transparency, we upheld our 80% threshold, and have been clear in our reporting of the included statements where consensus was between 75–79%.

## Implications

Many of our expert participants in the e-Delphi shared their reservations on the clinical and critical tone of several of the e-Delphi items. Whilst participants agreed there is a need to identify a parent's strengths and needs, and to show and celebrate progress, there was strong suggestion for a more family-centred, collaborative, and non-judgemental approach within the assessment. Consideration of family readiness, and of the parents' emotional well-being should always be in the foreground for any PCI assessment [16]. Establishing a balanced parent/professional partnership based on compassion, openness, and trust can result in a family who are more likely to positively receive the assessment and ongoing support, and then begin to build

their own efficacy [16]. We therefore remind the reader that the outcome of this e-Delphi was to gain consensus on the core content of a new assessment tool and approaches to be used. We would not advise the use of this content in its current state as there is another important phase to follow for the assessment's development: coproduction with parents of deaf children and hearing and deaf professionals.

Since sharing the findings with our PPI group, parents of deaf children have shared concerns over professionals using the terms 'assessment' and 'goal' in families' homes. Thus, co-production work will address issues such as language-use and develop terminology and an approach for observing PCI and evidencing skill-change that is non-judgemental, and family-centred while remaining evidence-based. As can be seen in S1 Table, reworded statements were longer in almost every instance, suggesting that contextual information and/or conditional elements such as language use were important factors in increasing consensus. There will therefore be longer statements within the assessment tool.

A pilot version of the Early Parent Interaction in Deafness (EPID) Tool and manual will be created in 2024. Once coproduction phases are complete, the EPID will be piloted in early years services in England so as to test the tool's psychometrics (i.e., reliability, validity, and responsiveness). We will also look at the tool's impact on parental knowledge, self-efficacy and on professionals' competence in appraising PCI in a supportive, strengths-based way. Following EPID training of a wider group of professionals, the clinical utility of the EPID and the extent and quality of its use will be investigated. This will include reviewing the acceptability, appropriateness, applicability, feasibility, adoption, and fidelity of the EPID in different services and cultural and language contexts. Data will be gathered through observation, feedback loops and surveys. We invite professionals interested in trialling the tool, both in the UK and internationally, to contact the lead author, but acknowledge that the tool may not be universally applicable in its pilot form. For some global contexts, more situational research and consideration is required before implementing a video-based tool that is grounded in the western view of language acquisition.

Whilst the assessment tool is still in development at this point, the outcomes of this study provide practitioners, academics, educators, and parents with a list of the internationally agreed, evidence-based, parent behaviours important for PCI where the child is deaf and aged 0–3. There are also some agreed recommendations on how to achieve a best practice PCI assessment; these will assist the coproduction phase.

## Conclusion

This study recruited 83 experts in the field of deafness to agree on the core content of a tool for PCI assessment for use with parents and deaf infants. A large number of parent behaviours reached consensus including parental sensitivity, positive affect, responsivity, maintaining joint engagement, and language facilitation techniques to ensure language is accessible and / or enriched. Some recommendations were also agreed upon for how a professional might approach an assessment of PCI. These included involving and educating the parent within the review of the assessment and being collaborative about next steps (i.e., discussing and jointly deciding upon areas to improve or adapt within their everyday routines). Developing e-Delphi statements into a PCI tool will support parents and professionals to effectively identify strengths within a parent's interactions with their deaf child, leading to a more informed, empowered parent who has greater impact on their deaf child's language.

## Supporting information

**S1 Table. Reworded statements from round 1 used in round 2.**
(DOCX)

**S2 Table. Number of statements per category achieving consensus.**
(DOCX)

**S1 File. Open text responses - additional needs.**
(DOCX)

**S2 File. Open text responses - multilingual families.**
(DOCX)

## Acknowledgments

We would like to thank all expert participants for their expertise and time in completing the e-Delphi and contributing towards the development of the assessment tool. Participants included: Alison Kendall, Dr Amy Lederberg, Alison Barley, Amanda Tivey, Amy Stephens, Anna Hughes, Ann East, Dr Alex Quittner, Dr Arlene Stredler Brown, Aya Oyama, Dr Bencie Woll, Dr Beatrijs Wille, Dr Patricia Elizabeth Spencer, Caroline Murphy, Dr Cathy Carotta, Charlotte Emery, Dr Emelie Cramér-Wolrath, Dr Daniel Holzinger, Dr Deborah James, Dr Derek Michael Houston, Dr Evelien Dirks, Dr Elaine Gale, Elizabeth Roche, Ellen Swann, Emma Mottram, Dr Gwen Carr, Helen O'Donnell, Hee Han Hui, Helene Elia, Dr Heidi J. Evans, Jacqueline Watton, Jane Fisher, Janice McWalter, Jayne Ramirez Inscoe, Jayne Langdale, Jane Gallacher, Jo Coote, Jean McAllister, Dr Joanna Hoskin, Julia Hollier, Julie Manning, Karen Reichmuth, Kathryn Reece, Katy Liriano, Kate Dixon, Dr Kerstin Watson Falkman, Kirsti Mackay, Lois Hatfield, Dr Lynne Sanford Koester, Dr Meghana Wadnerkar Kamble, Maria Nicastri, Marsha Locke, Mary Kean, Dr Mary Pat Moeller, Hilary Ann Dumbrill, Noel Kenely, Dr Patrizia Mancini, Dr Reinhild Glanemann, Dr Rachel O'Neill, Rachel Ward, Rhian Gardner, Sabrina Anderson, Sandra David, Sarah Allen, Dr Sophie Ambrose, Dr Susan Louise Tasker, Suzanne Harrigan, Teresa Quail, Vicki Lorriman Hughes, Wendy Martin, and Yvonne Seebens.

We would also like to give special mention and thanks to the research involvement group of deaf and hearing parents and professionals, who have been, and continue to be, vital in the development of this work: Alison Miles, Devi Krishnamoorthy, Dr Evelyne Mercure, Hannah Lumby, Husna Begum, Julie Hare, Julie Hughes, Juliet Viney, Karin Schamroth, Katerina Giachritsi, Lisa Smith, Martine Monksfield, Michelle Rayne, Paria Moghaddar, Sabina Iqbal, Tarryn Jacobs, Tina Wakefield, and Yasmena Waris.

## Author Contributions

**Conceptualization:** Martina Curtin, Madeline Cruice, Rosalind Herman.

**Data curation:** Martina Curtin.

**Formal analysis:** Martina Curtin, Madeline Cruice, Rosalind Herman.

**Funding acquisition:** Martina Curtin, Madeline Cruice, Gary Morgan, Rosalind Herman.

**Investigation:** Martina Curtin, Madeline Cruice, Rosalind Herman.

**Methodology:** Martina Curtin, Madeline Cruice, Gary Morgan, Rosalind Herman.

**Project administration:** Martina Curtin, Rosalind Herman.

**Resources:** Martina Curtin.

**Software:** Martina Curtin.

**Supervision:** Madeline Cruice, Gary Morgan, Rosalind Herman.

**Validation:** Martina Curtin, Madeline Cruice, Rosalind Herman.

**Writing – original draft:** Martina Curtin.

**Writing – review & editing:** Martina Curtin, Madeline Cruice, Gary Morgan, Rosalind Herman.

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
