## [Decision Letter · Decision Letter 0]

19 Jan 2024

PONE-D-23-43233Assessing parent-child interaction with deaf and hard of hearing infants aged 0-3 years: An international multi-professional e-Delphi.PLOS ONE Thank you for submitting your manuscript to PLOS ONE. After careful consideration, we feel that it has merit but does not fully meet PLOS ONE’s publication criteria as it currently stands. Therefore, we invite you to submit a revised version of the manuscript that addresses the points raised during the review process.

We look forward to receiving your revised manuscript.

Kind regards,

Ateya Megahed Ibrahim El-eglany

Academic Editor

PLOS ONE

Journal Requirements:

2. Please provide the full ethics committee approval name in the main document.

Additional Editor Comments (if provided):

Dear Author,

I want to commend you on the meticulous effort you've put into addressing the reviewer's comments on your manuscript, particularly regarding the research design. The application of the e-Delphi method for developing consensus among experts is indeed commendable. Your detailed response to the reviewer's concern about the potential biases inherent in Delphi studies and the self-selecting nature of participants demonstrates a thorough understanding of the methodological nuances. However, I would encourage you to consider a more in-depth discussion of these limitations in the revised manuscript, exploring how they might influence the study's outcomes. By acknowledging and addressing these limitations head-on, you can enhance the transparency and credibility of your research.

The reviewer also raised a pertinent point about participant selection and diversity. Your inclusion of an international and multi-professional panel is a notable strength of your study. To build upon this, providing additional details on how the diversity of the panel, in terms of geographic, cultural, and professional backgrounds, may have influenced the findings would offer valuable insights. This could potentially strengthen the generalizability of your results and contribute to a more nuanced understanding of the implications of your study.

Furthermore, I appreciate your attention to the data analysis process and the effective presentation of consensus data. To elevate this aspect, I suggest delving deeper into the analysis by discussing how divergent views were handled and exploring the implications of any significant disagreements among experts. This additional layer of analysis would contribute to a more comprehensive understanding of the nuances within your findings.

In terms of practical application, your manuscript could benefit from a more detailed discussion on how professionals can practically apply the findings in diverse settings. Providing specific guidelines or recommendations for practitioners using the assessment tool would not only add practical value but also enhance the potential impact of your research.

I also encourage you to expand on the future research directions. While you have mentioned potential avenues for further study, a more detailed exploration of how the assessment tool could be tested, refined, and implemented in real-world settings would be beneficial. Discussing pilot studies, feedback loops from practitioners, and adaptations for different cultural contexts could provide a roadmap for future researchers interested in building upon your work.

Lastly, your thorough consideration of ethical approval and patient involvement is commendable. However, the reviewer suggests a deeper discussion on the ethical considerations of implementing such tools in diverse cultural and linguistic settings. Expanding on this aspect would contribute to a more robust ethical framework for your study.

Overall, your manuscript shows great promise, and I appreciate your dedication to refining it. I believe that by incorporating these suggestions, you can further strengthen the methodological rigor, applicability, and impact of your research.

Reviewers' comments:

Reviewer's Responses to Questions

**Comments to the Author**

1. Is the manuscript technically sound, and do the data support the conclusions?

Reviewer #1: Partly

Reviewer #2: Yes

2. Has the statistical analysis been performed appropriately and rigorously? 

Reviewer #1: Yes

Reviewer #2: Yes

3. Have the authors made all data underlying the findings in their manuscript fully available?

Reviewer #1: Yes

Reviewer #2: Yes

4. Is the manuscript presented in an intelligible fashion and written in standard English?

Reviewer #1: Yes

Reviewer #2: Yes

5. Review Comments to the Author

Reviewer #1: Research Design: The use of the e-Delphi method for developing consensus among experts is commendable. However, there could be a more in-depth discussion of the limitations inherent in Delphi studies, such as potential biases due to the self-selecting nature of participants or the influence of the initial survey design on the outcomes.

Participant Selection and Diversity: The study's strength lies in its international and multi-professional panel. However, it would be beneficial to provide more details on how the panel's diversity (in terms of geographic, cultural, and professional background) may have influenced the findings.

Data Analysis: While the article presents the consensus data effectively, a more comprehensive analysis, including how divergent views were handled and the implications of any significant disagreements among experts, would be insightful.

Practical Application: The article could benefit from a more detailed discussion on how the findings can be practically applied in various settings. Specific guidelines or recommendations for professionals using the assessment tool in diverse contexts would add value.

Future Research Directions: While future directions are mentioned, a more detailed exploration of how this tool could be tested, refined, and implemented in real-world settings would be useful. This could include pilot studies, feedback loops from practitioners, and adaptation for different cultural contexts.

Ethical Considerations: The ethical approval and patient involvement are well-addressed. It would be advantageous to further discuss the ethical considerations of implementing such tools in diverse cultural and linguistic settings.

Limitations: The article could provide a deeper exploration of its limitations, particularly regarding the generalizability of the findings across different cultures and languages, and how these limitations might affect the tool's applicability in various global contexts.

Conclusion: The conclusion effectively summarizes the findings but could further emphasize the practical implications and potential impact of this research on early intervention programs for deaf children.

Reviewer #2: Thank you for your valuable work.

Your work is valuable, The paper is well written. Few grammar mistakes were found however it is not significant and not affected the manuscript generally. The discussion was very good and interesting.

6. PLOS authors have the option to publish the peer review history of their article (what does this mean?). If published, this will include your full peer review and any attached files.

Reviewer #1: **Yes: **Mostafa Shaban

Reviewer #2: **Yes: **Walid Shaban Abdella

---

## [Author Response · Author response to Decision Letter 0]

1 Mar 2024

FAO: Academic Editor, PLOS ONE

Re: ‘Assessing parent-child interaction with deaf and hard of hearing infants aged 0-3 years: An international multi-professional e-Delphi’ (PONE-D-23-43233)

1st March 2024

Dear Dr Ateya Megahed Ibrahim El-eglany,

Thank you for your time and careful consideration of our paper for PLOS One and for gaining two very helpful reviews from Dr Mostafa Shaban, and Dr Walid Shaban Abdella. We appreciate your encouraging feedback on our work having merit and showing promise.

We also thank both reviewers for their strong appraisals of our paper, noting our technically sound manuscript, appropriate statistical analysis, valid conclusions, and their mention of valuable and interesting work. We are also very grateful for their positively worded, and constructive suggestions for improving the paper. Along with our responses below, the re-submitted manuscript addresses your and the reviewers’ comments and we feel a stronger paper has been developed as a result. 

Unfortunately, I could not update the order of my files within the portal, so the manuscript with highlighted revisions is at the very end of the whole PDF.

We wanted to note that this is our first re-submission of our manuscript, and therefore our first opportunity at responding to reviewers’ comments.

Thank you again for the opportunity to strengthen our paper. We look forward to hearing from you. 

With best wishes, 

Martina Curtin

Corresponding Author

Clinical Lead Specialist Speech and Language Therapist (Deaf Children and Young People) &

NIHR Clinical Doctorate Research Fellow

City, University of London: martina.curtin.1@city.ac.uk

Homerton University Hospital NHS Trust: martina.curtin@nhs.net

Responses to Peer Review: 

Point 1 

Additional Editor Comments:

I want to commend you on the meticulous effort you've put into addressing the reviewer's comments on your manuscript, particularly regarding the research design. The application of the e-Delphi method for developing consensus among experts is indeed commendable. Your detailed response to the reviewer's concern about the potential biases inherent in Delphi studies and the self-selecting nature of participants demonstrates a thorough understanding of the methodological nuances.

Authorship Team Response: As noted in our letter above, this is our first re-submission and therefore our first attempt at addressing reviewer’s comments. 

Point 2

Additional Editor Comments: I would encourage you to consider a more in-depth discussion of these limitations in the revised manuscript, exploring how they might influence the study's outcomes. By acknowledging and addressing these limitations head-on, you can enhance the transparency and credibility of your research.

Reviewer 1’s Linked Comment:

Research Design: The use of the e-Delphi method for developing consensus among experts is commendable. However, there could be a more in-depth discussion of the limitations inherent in Delphi studies, such as potential biases due to the self-selecting nature of participants or the influence of the initial survey design on the outcomes.

Authorship Team Response: Thank you. Changes made. We have added the following paragraph to Limitations, p.43-44:

‘It is important to note the inherent limitations within Delphi studies, that are also relevant to this research project. Firstly, how survey items are designed, such as their use of abstract language and sentence length, has a proven influence on Delphi outcomes (62). The piloting phase with a range of professional group addressed this issue in our study to some degree. Secondly, participants were self-selecting individuals utilising their opinion to answer survey items. Individual agency, values, experiences, interpretations, agendas, and social and political climates underscore the answers given within Delphis (63). We invite the reader to hold these limitations in mind when interpreting the results.’

Point 3

Additional Editor Comments: The reviewer also raised a pertinent point about participant selection and diversity. Your inclusion of an international and multi-professional panel is a notable strength of your study. To build upon this, providing additional details on how the diversity of the panel, in terms of geographic, cultural, and professional backgrounds, may have influenced the findings would offer valuable insights. This could potentially strengthen the generalizability of your results and contribute to a more nuanced understanding of the implications of your study.

Reviewer 1’s Linked Comment: Participant Selection and Diversity: The study's strength lies in its international and multi-professional panel. However, it would be beneficial to provide more details on how the panel's diversity (in terms of geographic, cultural, and professional background) may have influenced the findings.

Authorship Team Response: Thank you. Changes made. We have added the following to Limitations, pg. 43: 

‘Whilst there are merits in the highly experienced, international, and multi-professional expert participants, we did not collect information on the ethnicities or cultural backgrounds of those recruited. Despite not knowing the cultural or linguistic backgrounds of participants, nearly all participants were recruited from western, educated, industrialised, rich, and democratic countries (WEIRD). This likely means that their research, practice and/or experiences will be grounded in a mainstream view of language acquisition for deaf infants, summarised in a systematic review (14). This may also explain low instances of divergence within the study. Some caution therefore must be taken when interpreting these findings more globally.’ 

Point 4

Additional Editor Comments:

Furthermore, I appreciate your attention to the data analysis process and the effective presentation of consensus data. To elevate this aspect, I suggest delving deeper into the analysis by discussing how divergent views were handled and exploring the implications of any significant disagreements among experts. This additional layer of analysis would contribute to a more comprehensive understanding of the nuances within your findings.

Reviewer 1’s Linked Comment:

Data Analysis: While the article presents the consensus data effectively, a more comprehensive analysis, including how divergent views were handled and the implications of any significant disagreements among experts, would be insightful.

Authorship Team Response:

Thank you. Changes made. Because of this useful suggestion, we delved deeper into the evidence base for consensus indices in Delphis and note Birko et al (https://journals.plos.org/plosone/article?id=10.1371/journal.pone.0135162) list IQRs as one of the nine important indices. We were already using IQRs to present our data, but we have further highlighted IQR use throughout the paper (i.e. IQR of <1 for consensus and IQR>1 as divergence). This refinement has been added to Consensus (on p9).

To highlight divergence, we have added asterisks to items with divergence in tables 3 and 4 and added the following to Round 1 Results ‘Review and Rewording’ on pg 20: 

‘There were seven PBs and six AAs where more divergent responses were present (i.e., IQRs >1). These are marked with an asterisk in the tables below. All of the statements, but particularly those with dissenting judgements, were carefully scrutinised using participants’ qualitative responses to aid the rewording of items. The review and rewording process resulted in less divergence in round 2.’

To follow through with the divergent findings, we then added this sentence to Round 2 findings for PBs: (p.23) 

‘This included six of the seven previously divergent PBs (i.e., IQR>1)’.

And this sentence to Round 2 findings for AAs (p.33): 

‘This included three of the six previously divergent AAs (i.e., IQR>1)’.

We then used SPSS to look more closely at the divergent items (8 PBs and 8 AAs across rounds 1 and 2) to see if there were any differences between how professional groups or participants in different geographical areas rated these items. We did not find any significant differences and so we propose this is because of our quite similar participants as noted in our new section in Limitations, p.43: 

‘This likely means that their research, practice and/or experiences will be grounded in a mainstream view of language acquisition for deaf infants, summarised in a systematic review (14). This may also explain low instances of divergence within the study.’

Point 5

Additional Editor Comments:

In terms of practical application, your manuscript could benefit from a more detailed discussion on how professionals can practically apply the findings in diverse settings. Providing specific guidelines or recommendations for practitioners using the assessment tool would not only add practical value but also enhance the potential impact of your research.

Reviewer 1’s Linked Comment:

Practical Application: The article could benefit from a more detailed discussion on how the findings can be practically applied in various settings. Specific guidelines or recommendations for professionals using the assessment tool in diverse contexts would add value.

Authorship Team Response: 

Thank you for this. The paper has benefitted after reflecting and further clarifying this point. Changes made on pages 45-46. 

We have re-read the manuscript and see that it is possible we may have led the reader into thinking the PCI assessment tool is finished, i.e. the outcomes of this e-Delphi is the new assessment. The paper describes how a Delphi study has produced the core content for an assessment tool, but we have more work to do before we produce the finished tool. An important next step is for the professional approach within the assessment to be coproduced with parents of deaf children. We have been sure not to ‘oversell’ the findings of this paper and have reviewed all these instances to ensure wording is accurately reflecting the project’s outcomes.

We have also added the following to the first paragraph of Implications on page 45:

‘Many of our expert participants in the e-Delphi shared their reservations on the clinical and critical tone of many of the e-Delphi items. Whilst many agreed there is a need to identify a parent’s strengths and needs, and to show and celebrate progress, there was strong suggestion for a more family-centred, collaborative, and non-judgemental approach within the assessment. Consideration of family readiness, and of the parents’ emotional well-being should always be in the foreground and set the pace for any PCI assessment (65). Establishing a balanced parent/professional partnership based on compassion, openness, and trust can result in a family who are more likely to positively receive the assessment and ongoing support, and begin to build their own efficacy (65). We therefore remind the reader that the outcome of this e-Delphi was to gain consensus on the core content of a new assessment tool and approaches to be used. We would not advise the use of this content in its current state as there is another important phase to follow for the assessment’s development: coproduction with parents of deaf children and hearing and deaf professionals.’

And we have added this as a final paragraph to the same, Implications section on page 46: 

‘Whilst the assessment tool is still in development at this point, the outcomes of this study provide practitioners, academics, educators, and parents with a list of the internationally agreed, evidence-based, parent behaviours important for PCI where the child is deaf and aged 0-3. There are also some agreed recommendations on how to achieve a best practice PCI assessment; these will assist the coproduction phase’.

Point 6

Additional Editor Comments:

I also encourage you to expand on the future research directions. While you have mentioned potential avenues for further study, a more detailed exploration of how the assessment tool could be tested, refined, and implemented in real-world settings would be beneficial. Discussing pilot studies, feedback loops from practitioners, and adaptations for different cultural contexts could provide a roadmap for future researchers interested in building upon your work.

Reviewer 1’s Linked Comment:

Future Research Directions: While future directions are mentioned, a more detailed exploration of how this tool could be tested, refined, and implemented in real-world settings would be useful. This could include pilot studies, feedback loops from practitioners, and adaptation for different cultural contexts.

Authorship Team Response: 

Thank you. Changes made. We have added these sentences to the penultimate paragraph of the Implications section on pages 45-46: 

‘Once coproduction phases are complete, the EPID will be piloted in early years services in England so as to test the tool’s psychometrics (i.e., reliability, validity, and responsiveness). We will also look at the tool’s impact on parental knowledge, self-efficacy and on professionals’ competence in appraising PCI in a supportive, strengths-based way. Following EPID training of a wider group of professionals, the clinical utility of the EPID and the extent and quality of its use will be investigated. This will include reviewing the acceptability, appropriateness, applicability, feasibility, adoption, and fidelity of the EPID in different services and cultural and language contexts. Data will be gathered through observation, feedback loops and surveys. We invite professionals interested in trialling the tool, both in the UK and internationally, to contact the lead author, but acknowledge that the tool may not be universally applicable in its pilot form. For some global contexts, more situational research and consideration is required before implementing a video-based tool that is grounded in the western view of language acquisition.’ 

Point 7

Additional Editor Comments:

Your thorough consideration of ethical approval and patient involvement is commendable. However, the reviewer suggests a deeper discussion on the ethical considerations of implementing such tools in diverse cultural and linguistic settings. Expanding on this aspect would contribute to a more robust ethical framework for your study.

Reviewer 1’s Linked Comment:

Ethical Considerations: The ethical approval and patient involvement are well-addressed. It would be advantageous to further discuss the ethical considerations of implementing such tools in diverse cultural and linguistic settings.

Reviewer 1’s Linked Comment:

Limitations: The article could provide a deeper exploration of its limitations, particularly regarding the generalizability of the findings across different cultures and languages, and how these limitations might affect the tool's applicability in various global contexts.

Authorship Team Response:

Thank you. Changes made. We feel that the following additions to our paper (already outlined above in points 3 and 6 have addressed these points.

Limitations, p43 – point 3 additions: 'Whilst there are merits in the highly experienced, international, and multi-professional expert participants, we did not collect information on the ethnicities or cultural backgrounds of those recruited. Despite not knowing the cultural or linguistic backgrounds of participants, nearly all participants were recruited from western, educated, industrialised, rich, and democratic countries (WEIRD). This likely means that their research, practice and/or experiences will be grounded in a mainstream view of language acquisition for deaf infants, summarised in a systematic review (16). Caution therefore must be taken when interpreting and/or utilising these findings with diverse cultural and linguistic contexts globally.’

Implications, p46 – point 6 addition: ‘We invite professionals interested in trialling the tool, both in the UK and internationally, to contact the lead author, but acknowledge that the tool may not be universally applicable in its pilot form. For some global contexts, more situational research and consideration is required before implementing a video-based tool that is grounded in the wester

---

## [Decision Letter · Decision Letter 1]

22 Mar 2024

Assessing parent-child interaction with deaf and hard of hearing infants aged 0-3 years: An international multi-professional e-Delphi.

PONE-D-23-43233R1

Dear Dr. Curtin 

We’re pleased to inform you that your manuscript has been judged scientifically suitable for publication and will be formally accepted for publication once it meets all outstanding technical requirements.

Kind regards,

Ateya Megahed Ibrahim El-eglany

Academic Editor

PLOS ONE

Additional Editor Comments (optional):

Reviewers' comments:

Reviewer's Responses to Questions

**Comments to the Author**

1. If the authors have adequately addressed your comments raised in a previous round of review and you feel that this manuscript is now acceptable for publication, you may indicate that here to bypass the “Comments to the Author” section, enter your conflict of interest statement in the “Confidential to Editor” section, and submit your "Accept" recommendation.

Reviewer #1: All comments have been addressed

Reviewer #3: All comments have been addressed

2. Is the manuscript technically sound, and do the data support the conclusions?

Reviewer #1: Yes

Reviewer #3: Yes

3. Has the statistical analysis been performed appropriately and rigorously? 

Reviewer #1: Yes

Reviewer #3: Yes

4. Have the authors made all data underlying the findings in their manuscript fully available?

Reviewer #1: Yes

Reviewer #3: Yes

5. Is the manuscript presented in an intelligible fashion and written in standard English?

Reviewer #1: Yes

Reviewer #3: Yes

6. Review Comments to the Author

Reviewer #1: Dear author

thank you bfor your revision and effort you put into modifying the paper

i accept this paper without comments

Reviewer #3: I hope this message finds you well. I wanted to take a moment to express my sincere appreciation for your outstanding work on the manuscript titled "Assessing Parent-Child Interaction with Deaf and Hard of Hearing Infants Aged 0-3 Years: An International Multi-Professional e-Delphi." As a reviewer, I have had the privilege of assessing the content, structure, and methodology of your research, and I am thoroughly impressed by the quality of your contribution.

Your dedication to exploring the critical topic of parent-child interaction with deaf and hard of hearing infants within the age range of 0-3 years shines through in your meticulous approach and comprehensive analysis. Your utilization of the e-Delphi method adds a unique dimension to the study, facilitating an international perspective and multi-professional insights that enrich the findings.

Furthermore, I commend your responsiveness and diligence in addressing all the required modifications suggested during the review process. Your willingness to engage with feedback and refine your work demonstrates your commitment to academic rigor and the advancement of knowledge in this field.

The significance of your research cannot be overstated, as it not only contributes to our understanding of parent-child interaction dynamics but also has the potential to inform interventions and support systems for families with deaf and hard of hearing infants. Your work holds immense value for both scholarly discourse and practical applications in clinical and educational settings.

Once again, I extend my gratitude for your exemplary efforts and congratulate you on the successful completion of this manuscript. It has been a pleasure to engage with your work, and I look forward to seeing its impact unfold within the academic community and beyond.

Best regards,

7. PLOS authors have the option to publish the peer review history of their article (what does this mean?). If published, this will include your full peer review and any attached files.

Reviewer #1: **Yes: **Mostafa Shaban

Reviewer #3: **Yes: **Ateya Megahed Ibrahim
